# Palaeogeographic regulation of glacial events during the Cretaceous supergreenhouse

Jean-Baptiste Ladant[1] & Yannick Donnadieu[1,2]

The historical view of a uniformly warm Cretaceous is being increasingly challenged by the accumulation of new data hinting at the possibility of glacial events, even during the Cenomanian–Turonian ($\sim$95 Myr ago), the warmest interval of the Cretaceous. Here we show that the palaeogeography typifying the Cenomanian–Turonian renders the Earth System resilient to glaciation with no perennial ice accumulation occurring under prescribed $CO_2$ levels as low as 420 p.p.m. Conversely, late Aptian ($\sim$115 Myr ago) and Maastrichtian ($\sim$70 Myr ago) continental configurations set the stage for cooler climatic conditions, favouring possible inception of Antarctic ice sheets under $CO_2$ concentrations, respectively, about 400 and 300 p.p.m. higher than for the Cenomanian–Turonian. Our simulations notably emphasize that palaeogeography can crucially impact global climate by modulating the $CO_2$ threshold for ice sheet inception and make the possibility of glacial events during the Cenomanian–Turonian unlikely.

[1] Laboratoire des Sciences du Climat et de l'Environnement, LSCE-IPSL, CEA-CNRS-UVSQ, Université Paris-Saclay, 91191 Gif-sur-Yvette, France.
[2] Aix-Marseille Université, CNRS, IRD, CEREGE, UM34, 13545 Aix-en-Provence, France. Correspondence and requests for materials should be addressed to J.-B.L. (email: jean-baptiste.ladant@lsce.ipsl.fr).

Estimates of Cretaceous climates reveal warmer inland[1,2] and oceanic temperatures[3–6] relative to today, possibly reaching up to more than 5 °C warmer in the low latitudes[3,4] and more than 15 °C warmer in the high latitudes[7]. This led to reduced meridional temperature gradients[8], as indicated for instance by evidence for ectothermic crocodilian species poleward of 70° N (ref. 9) or fossil woods on the Antarctic Peninsula[2]. This broad view of Cretaceous warmth has been challenged in recent decades by the emergence of a substantial amount of data suggesting greater variability in the climate of the Cretaceous. In particular, geochemical proxies point to long-term changes in temperature with a warming trend from the late Aptian to the Cenomanian–Turonian followed by a cooling trend through to the Maastrichtian[10,11], while several pieces of evidence[12], including sea-level reconstructions[13], oxygen isotope excursions[14] and palynomorph records[15], converge to suggest that small to intermediate polar ice sheets may have periodically developed, specifically during the aforementioned geological stages. Cold high-latitude conditions have notably been reported for the late Aptian and Maastrichtian[15–18], making plausible the presence of polar ice sheets despite the absence of direct evidence of glaciations. On the other hand, arguments in favour of glacial episodes during the Cenomanian–Turonian rely on the interpretation of oxygen isotope records[14,19] and sequence stratigraphy[20,21], although the validity of these proxies as recorders of Cenomanian–Turonian glacioeustasy is vividly debated[22]. The downward revision of atmospheric pCO$_2$ estimates for the mid-Cretaceous[23–25], to levels possibly below those inferred from previous modelling studies[26,27] and proxy records[28,29] at the time of the major Eocene–Oligocene (EO) Antarctic glaciation[30,31] (∼34 Myr ago), adds support to the possibility of Cenomanian–Turonian ice sheets.

In this study, through the use of a coupled set-up between climate and ice sheet models, we investigate the impact of palaeogeographic changes on the sensitivity of the Cretaceous climatic system to glacial inception. Many studies have indeed demonstrated that palaeogeographic changes could generate substantial global and/or regional climatic variations[32–34]. However, most studies that have coupled deep-time climate and ice sheets have focused on the Cenozoic initiation of the Antarctic and Greenland glaciations[26,27]. Although some studies have looked back further in time with a focus on the Ordovician and on the Permo-Carboniferous glaciations[34,35] they remain sporadic. We apply here a coupled climate-ice sheet model on the Cretaceous. Our simulations highlight the role of palaeogeography in maintaining a warmer Cenomanian–Turonian relative to the late Aptian and Maastrichtian, thus supporting the absence of ice during the Cretaceous so-called Climatic Optimum.

## Results

**Experimental background.** We perform numerical experiments with three different palaeogeographies[36], representing the late Aptian, the Cenomanian–Turonian (hereafter, Turonian) and the early Maastrichtian (Supplementary Fig. 1). It is important to emphasize that uncertainties in Cretaceous palaeogeographic reconstructions and the model resolution preclude an exact correlation between a palaeogeography and a specific age. In particular, our Turonian palaeogeography could appropriately represent the mid-Cenomanian, but is referred to as Turonian for clarity. The same Antarctic topography is imposed in the three palaeogeographies to avoid local topographic biases (see Methods). The solar constant is adapted through time according to Crowley and Baum[37], and the orbital parameters of the Earth are designed to produce cold summer conditions over the

Southern high latitudes to favour the onset of ice sheets. Three different pCO$_2$ levels are prescribed (560, 840 and 1,120 p.p.m.) to cover the estimated range for the end of the Mesozoic[23–25,38–40], and other boundary conditions are kept to modern values. To evaluate the potential development of polar ice sheets, we use a relatively simple coupled climate-ice sheet methodology, validated against results from a more elaborated coupled method[26] on the Antarctic glaciation at the EO transition (Methods). Additional sensitivity experiments at other pCO$_2$ (between 280 and 750 p.p.m.) are then performed to better constrain the CO$_2$ threshold for glacial inception over Antarctica for each palaeogeography.

**Ice sheet sensitivity to cretaceous palaeogeography.** In our experiments, palaeogeographic reorganizations profoundly affect the distribution of land ice over the high latitudes of the Earth (Fig. 1 and Supplementary Figs 2 and 3). At 1,120 p.p.m., the climate is warm enough to prevent any significant ice growth. When the atmospheric pCO$_2$ is decreased to 840 p.p.m., small isolated ice caps develop over Antarctica's highest peaks in the Aptian configuration but their sizes are too modest to remain perennial (Methods). In contrast, a further drop to 560 p.p.m. allows a substantial amount of ice to accumulate over the Antarctic continent in the Aptian and Maastrichtian configurations while the Turonian Antarctic persists in an ice-free state (Fig. 1). In the following, we will thus put emphasis on the results from the 560 p.p.m. simulations to investigate the causes of the Turonian warmth that prevents the nucleation of ice over Antarctica. However, additional sensitivity simulations at other pCO$_2$ (650 and 750 p.p.m. for the Aptian and the Maastrichtian, and 280 and 420 p.p.m. for the Turonian) demonstrate that the Antarctic ice sheet CO$_2$ threshold is crossed, respectively, between 840 and 750 p.p.m. and between 750 and 650 p.p.m. for the Aptian and Maastrichtian configurations (Supplementary Figs 2 and 4). In contrast, the inception of an ice sheet over Antarctica in the Turonian configuration occurs for a CO$_2$ threshold between 420 and 280 p.p.m., that is, for CO$_2$ levels about 400 and 300 p.p.m. lower than the Aptian and Maastrichtian. We also perform sensitivity experiments to investigate if Northern Hemisphere ice sheets could develop in the Cretaceous world. Simulations with a cold boreal summer orbital configuration and a CO$_2$ concentration of 560 p.p.m. show a lower sensitivity of the Northern Hemisphere to glaciation with no accumulation of ice in any of the three palaeogeographies (Supplementary Fig. 3), as also demonstrated for the Cenozoic[27].

Antarctic ice sheet mass balance analyses for the 560 p.p.m. experiments show that the Aptian and Maastrichtian configurations exhibit large areas with positive annual mass balance, whereas the Turonian Antarctic does not (Supplementary Fig. 5). Such a change in mass balance is essentially driven either by a decrease of the annual snow accumulation or an increase in the annual ablation. In our simulations, the accumulation term is similar between the different palaeogeographies, whereas the Turonian experiences strong ablation over almost the whole Antarctic continent, indicating that summer temperatures drive the absence of ice in the Turonian (Supplementary Fig. 5). To confirm this assumption, we perform a sensitivity test consisting in two ice sheet experiments at 560 p.p.m. driven either by Turonian temperatures and Aptian precipitations or Aptian temperatures and Turonian precipitations (Supplementary Fig. 6). The accumulation of ice in the latter, in contrast to the first experiment, further demonstrates that Turonian summer temperatures are instrumental in preventing the inception of an ice sheet.

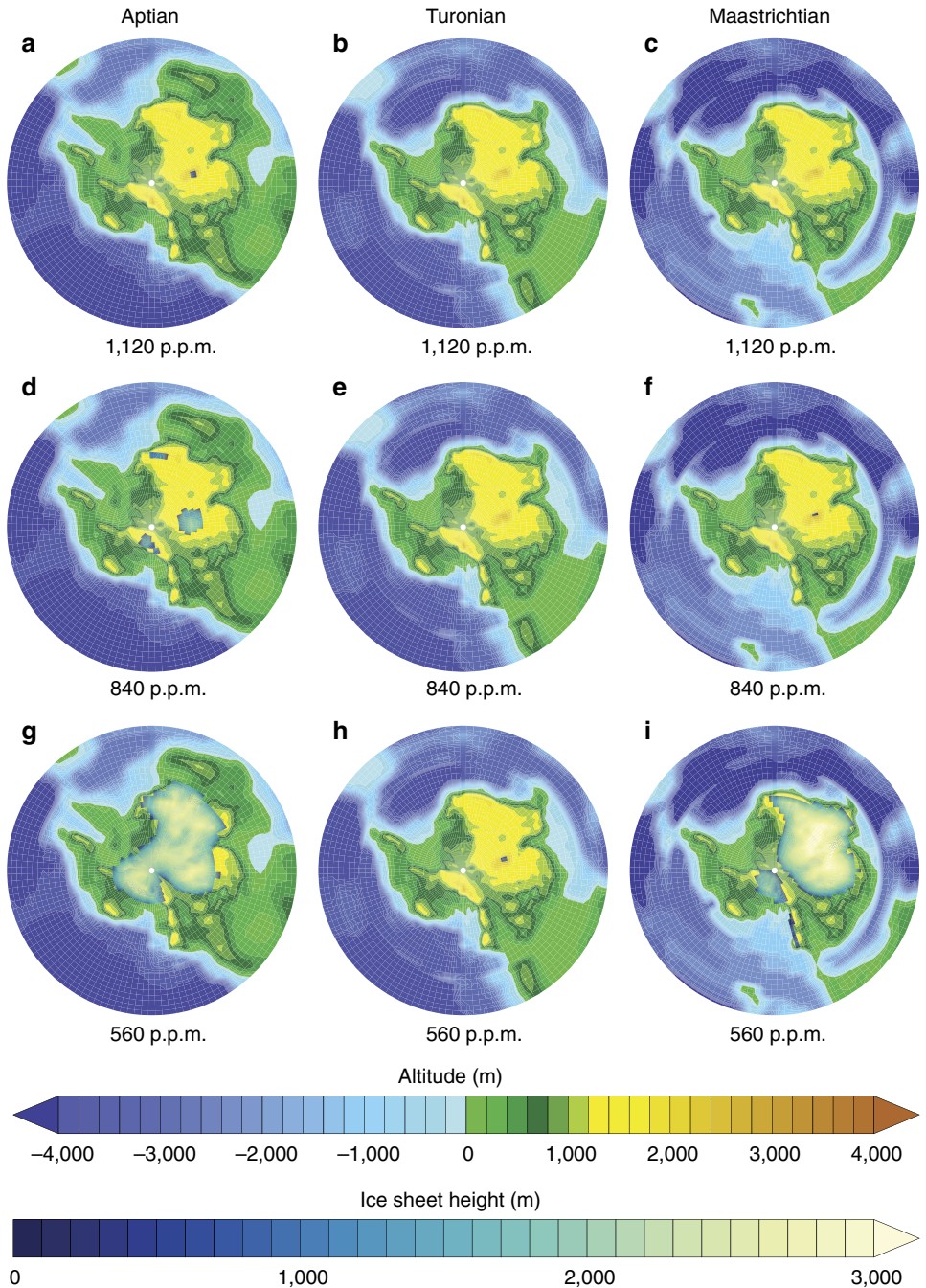

**Figure 1 | Simulated Antarctic ice sheet for each palaeogeography and different $CO_2$ concentrations.** (**a**,**d**,**g**) Aptian Antarctic ice sheet for atmospheric $CO_2$ concentrations of (**a**) 1,120, (**d**) 840 and (**g**) 560 p.p.m. after 10 kyr of integration of the Ice Sheet Model under a constant cold austral summer orbit. (**b**,**e**,**h**) Same for the Turonian. (**c**,**f**,**i**) Same for the Maastrichtian.

**Mechanisms of Turonian warmth.** Global mean calculations indeed demonstrate that the Turonian world is the warmest and the wettest irrespective of the $CO_2$ level (Supplementary Table 1). Turonian global mean annual temperature (MAT) and specific humidity ($Q$) at 500 hPa are consistently about 3 °C (0.5 g kg$^{-1}$) higher than Aptian MAT ($Q$) and about 2 °C (0.3 g kg$^{-1}$) higher than Maastrichtian MAT ($Q$). In particular, the global warming during the Turonian displays regionally non-uniform patterns, with a significant polar amplification and strong seasonal variations (Fig. 2).

A strong warming occurs over Turonian high latitudes relative to the Aptian and Maastrichtian. The largest temperature

differences are associated with continental areas becoming open ocean and vice versa. In the Northern Hemisphere, the majority of the warming occurs in winter and is related to the combination of a drastic decrease in sea-ice cover and of a more fragmented Turonian continental configuration, which acts to reduce the continentality effect because of the greater thermal capacity of the oceans (for example, ref. 41). This produces a temperature increase of more than 10 °C in the Arctic Ocean and over North America during the winter (Fig. 2c,d). In the Southern Hemisphere, local temperature differences reach more than 5 °C over most of the Antarctic interior between the Turonian and the Aptian regardless of the season and peak to about 10 °C during

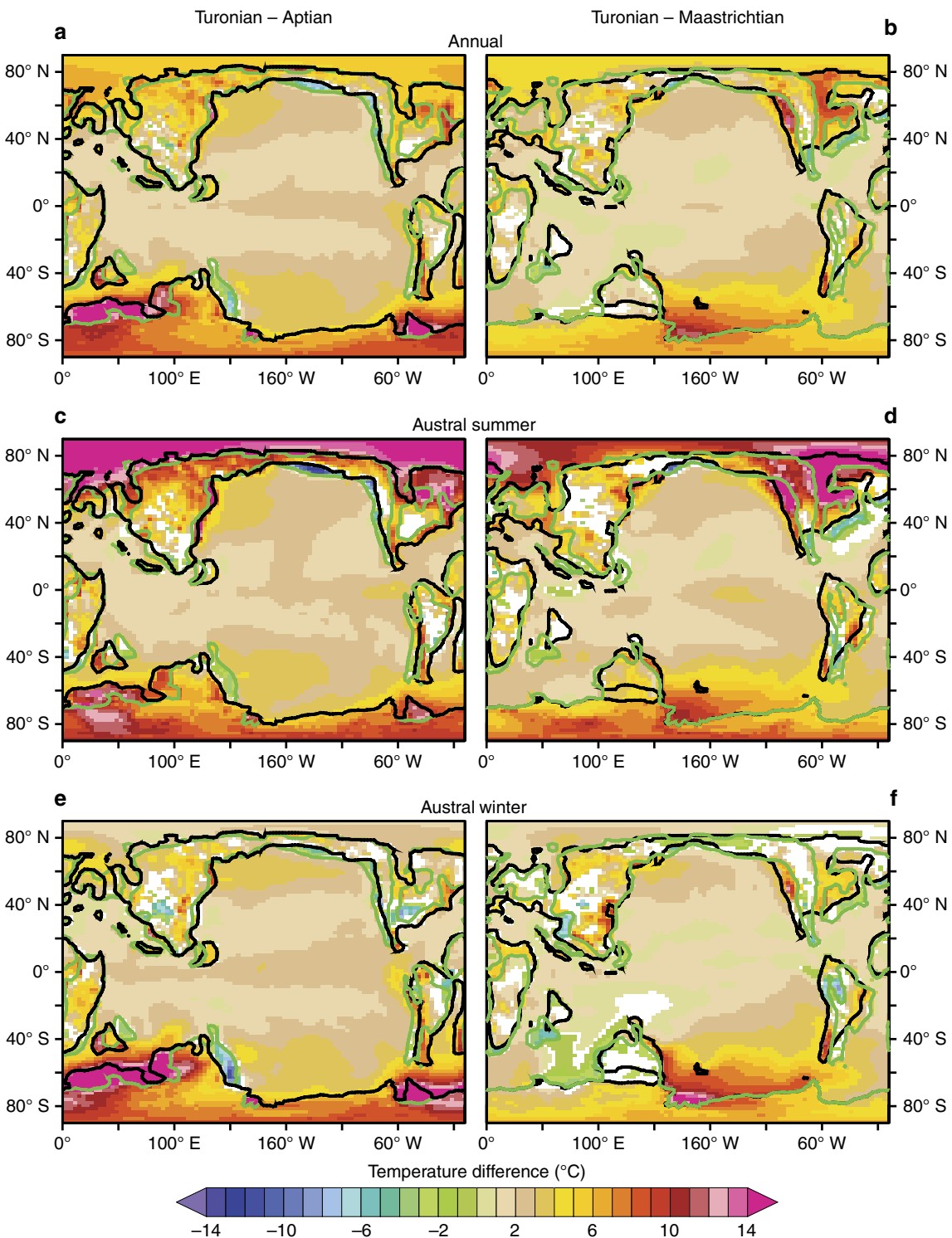

**Figure 2 | Annual and seasonal temperature difference between the Turonian and the Aptian or Maastrichtian.** (**a,c,e**) Statistically significant (at 95%, coloured cells) annual, austral summer and austral winter temperature (at 2 m) differences between the Turonian and the Aptian. Green (black) contours are the Turonian (Aptian) palaeo-shorelines. (**b,d,f**) Statistically significant (at 95%, coloured cells) annual, austral summer and austral winter temperature (at 2 m) differences between the Turonian and the Maastrichtian. Green (black) contours are the Turonian (Maastrichtian) palaeo-shorelines.

the summer (Fig. 2a,c,e). Similar temperature differences occur in the southern high latitudes between the Turonian and the Maastrichtian although of smaller amplitude (Fig. 2b,d,f). There is a significant ocean warming in the Atlantic and Indian sectors of the Southern Ocean during the Turonian relative to the Aptian. On the contrary, relative to the Maastrichtian, the Turonian

configuration displays a substantial warming in the Southern Pacific Ocean and a small cooling in the Southern Indian Ocean. These ocean changes are closely correlated to the onset or shutdown of convective mixing areas—interpreted as deep-water formation zones—in the Southern Ocean (Fig. 2 and Supplementary Fig. 7). Deep-water formation increases advection

of warm low-latitude surface waters, the radiative cooling of which (especially during winter) generates vertical mixing between these waters and the warmer subsurface. This efficiently prevents sea-ice formation and limits the cooling of the ocean at high latitude[42], resulting in strong warming anomalies. These differences in convective mixing zones are directly linked to palaeogeographic differences. First, the Aptian to Turonian ocean circulation changes are attributed to the opening of the equatorial Atlantic gateway between Africa and South America, which is a robust feature of the mid-Cretaceous palaeogeographic evolution (for example, the other Cretaceous reconstructions used in ref. 33). This opening indeed generates an export of warm and saline upper ocean waters into the South Atlantic, triggering deep-water formation and a strong ocean warming[43]. Second, a recent study has investigated the changes in ocean circulation and areas of deep-water formation between the Turonian and the Maastrichtian[44], including sensitivity experiments with different gateways configurations, and has attributed these changes to modifications of the South Atlantic hydrological cycle and of the Caribbean Seaway configuration.

The ocean circulation changes between the different Stages significantly impact the amount of heat transported poleward by the ocean (Supplementary Fig. 8a). The Turonian configuration indeed results in a larger southward extratropical export of heat via the ocean in spite of near-identical total heat transport. It has recently been proposed that this process would efficiently moisten the mid to high latitudes of the Southern Hemisphere through atmospheric convective adjustment[45]. We argue here that this mechanism, triggered by palaeogeographic reorganizations, can explain the absence of ice accumulation over Antarctica during the Turonian. During this stage, and relative to the Aptian and the Maastrichtian, convective precipitations in the southern mid and high latitudes effectively increase (Supplementary Fig. 8b) and a strong high-latitude summer warming occurs in conjunction with a substantial increase in upper troposphere and stratosphere water vapour (Fig. 3a,b, black and white contours). Although the specific humidity increases in the Turonian troposphere because of the global warming (Fig. 3a,b, white contours), the relative humidity changes follow a dipole pattern with a decrease in the lower and middle troposphere and a substantial increase in the upper troposphere and stratosphere (Fig. 3a,b, black contours), consistent with the idealized model of Rose and Ferreira[45]. The Turonian lower and middle atmosphere is thus undersaturated in water vapour compared with the Aptian and the Maastrichtian, whereas the upper atmosphere is oversaturated. The increased quantity of water contained in the air and the redistribution of the water vapour saturation leads to two main consequences. First, the larger amount of upper atmospheric water vapour drastically enhances the summer greenhouse effect over Antarctica during the Turonian (Fig. 3c). Second, the cloud distribution in summer is significantly affected by the different saturation state of the Turonian atmosphere in water vapour (Supplementary Fig. 9), leading to a decrease in low- and mid-altitude clouds and an increase in higher-altitude clouds. The reduction in low clouds severely enhances the amount of absorbed solar radiations in the Turonian mid to high latitudes of the Southern Hemisphere (Fig. 3d). On the contrary, the increase in high-altitude clouds only slightly impacts the summer greenhouse effect over the southern high latitudes (Fig. 3c). These changes thereby add up to produce the powerful warming observed over the Antarctic continent during the Turonian. In addition, the albedo over Antarctica also decreases because of the near complete snow melting (Supplementary Fig. 10), which further enhances the amount of solar radiations absorbed in the very high latitudes (70–90° S) and consequently the warming (Fig. 3d). To summarize, the Turonian Antarctica is maintained free of ice through the combined warming effects of increased upper tropospheric moisture, decreased low- and mid-altitude clouds and a lower albedo during the summer season.

## Discussion

Aside from the polar ice issue, which has never been investigated before with a three-dimensional ice sheet model, our numerical results exhibit analogous conclusions with respect to previous modelling work. Similar to other coupled studies[44,46], sites of deep-water formation (interpreted as areas of convective mixing) are located in the North Pacific and in the Southern Ocean. The Aptian–Turonian South Atlantic temperature rise and onset of deep-water formation has also been previously documented[43]. In addition, the Aptian to Maastrichtian shift in Southern Ocean convective mixing areas is also consistent with the onset of deep-water formation in the South Atlantic and Proto-Indian Ocean during the Late Cretaceous inferred from neodymium isotopes[47]. Only a few studies have attempted to compare different Middle–Late Cretaceous palaeogeographies using coupled models[32,33,44]. In line with data[4,11], our Aptian to Turonian warming seems a robust feature in coupled climate models as other studies reach similar conclusions[32,33] despite different boundary conditions, notably in terms of land/sea masks and topography/bathymetry. On the other hand, there is no consensus yet in coupled modelling studies regarding the Turonian to Maastrichtian cooling seen in the data[10,11,48]. In spite of slightly different boundary conditions, our results are in close agreement regarding the sites of deep-water formation and the Turonian to Maastrichtian cooling of Donnadieu et al.[44] (their Supplementary Figs 1, 2 and 7), whereas the simulations of Lunt et al.[33] shows a warmer Maastrichtian, possibly because of different modes of ocean circulation, as sites of deep-water formation are not located in the same areas, or because of different palaeogeographic reconstructions. Noting that our locations for deep-water formation sites are also consistent with other studies focussing only on the Late Cretaceous[46,49], these inter-model differences underline the need for future studies aiming at refining the Middle–Late Cretaceous climatic evolution. We nonetheless draw attention to the similarity between the temperature trend derived from our numerical simulations and the Middle–Late Cretaceous temperature trend derived from Exmouth Plateau[10] and Western Tethyan platform[11] $\delta^{18}O$ data, and this, without invoking any $CO_2$ variations. It thus suggests that palaeogeographic modifications can play a role in modulating global temperatures, even if uncertainties associated with temperature and $CO_2$ estimates forestall any firm conclusion to be drawn.

The presence of high-latitude ice sheets during the warm Cretaceous is a long lasting debate with numerous and considerable implications, regarding interpretations not only of past proxy data and climatic system[6,12,14,21] but also of Cretaceous faunal and floral polar ecosystems[50]. While lending further support to the plausibility of Aptian and Maastrichtian glaciations[12,51], our results demonstrate that tectonic movements efficiently act to thwart the development of glacial periods in the Cenomanian–Turonian by decreasing the $CO_2$ threshold for glacial inception by several hundreds of p.p.m. A recent palaeo-$CO_2$ estimates compilation[40], although affected by large uncertainties, tends to indicate similar, if not higher, Cenomanian-Turonian $CO_2$ concentrations relative to the Aptian and the Maastrichtian. Putting this line of evidence together with a substantially lower $CO_2$ threshold, our study defends the vision of an ice-free Cenomanian–Turonian, which has remained up to now an issue under debate[21,22,52].

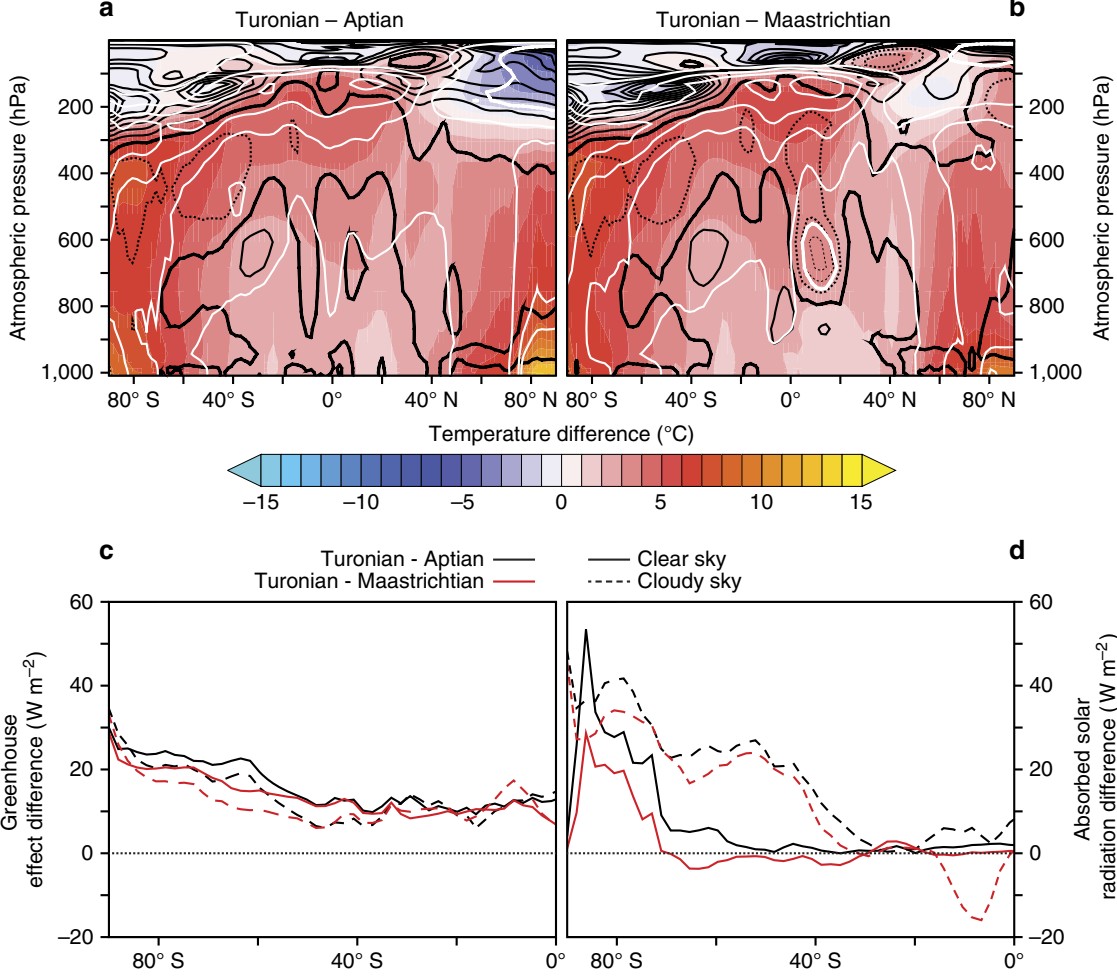

**Figure 3 | Atmospheric diagnostics of the Turonian warmth.** Zonally averaged austral summer temperature difference at 560 p.p.m. between (**a**) the Turonian and the Aptian and (**b**) the Turonian and the Maastrichtian. Black contours: change in relative humidity (contour interval 10%, zero contour bold, negative contours dashed). White contours: change in specific humidity (contour interval 25%, zero contour bold, negative contours dashed). (**c**) Summer greenhouse effect (GE) difference and (**d**) summer absorbed solar radiation (ASR) difference at 560 p.p.m. Black solid (dashed) lines are the clear (cloudy) sky difference between the Turonian and the Aptian. The difference between clear and cloudy sky shows the cloud forcing impact. Red lines are the difference between the Turonian and the Maastrichtian. GE is calculated as the difference between the surface upward longwave flux and the top-of-atmosphere outgoing longwave radiation while ASR is calculated at the top of atmosphere.

Cenomanian–Turonian Antarctic ice sheets both of continental-scale size and restricted to the interior of the continent have indeed been proposed on the basis of synchronous $\delta^{18}O$ excursions in planktic and benthic foraminifera with no evidence from a temperature contribution[14] and from proposed glacial lowstands on the New Jersey margin[20] and in Tethyan sections[21], the latter in conjunction with a bulk $\delta^{18}O$ excursion. In contrast, many other studies have found no evidence for glacial events during the Cenomanian–Turonian using oxygen isotopes records[6,52,53]. Our results corroborate the latter but new field studies are required to unambiguously settle the question of Cenomanian–Turonian ice sheets.

In conclusion, our findings suggest that the atmospheric $CO_2$ threshold leading to the inception of Antarctic ice sheets during the Middle–Late Cretaceous underwent significant changes of several hundreds of p.p.m. in response to palaeogeographic evolution. In particular, during the Aptian and Maastrichtian stages, this threshold is about 300–400 p.p.m. higher than during the Cenomanian–Turonian following complex ocean–atmosphere interactions that maintain significantly warmer (~5–10 °C) Antarctic summer temperatures in the

Cenomanian–Turonian configuration. Even though future work is needed to confirm these modelling results—as absolute $CO_2$ thresholds for glacial inception remain model-dependent—and to refine proxy-based estimates of $CO_2$ concentrations and their relative evolution through the Cretaceous, we suggest that Aptian and Maastrichtian ice sheets are indeed likely to have developed during times of favourable orbital forcing and atmospheric $CO_2$ levels. On the contrary, our study supports the vision of a super-hot, ice-free Cenomanian–Turonian, epitomizing the term of 'Climatic Optimum'.

## Methods

**Palaeogeographic reconstructions and boundary conditions.** Our palaeogeographic reconstructions are taken from ref. 36, which provides four palaeogeographic time slices for the Cretaceous (Supplementary Fig. 1). We modified these maps over Antarctica as its palaeotopography is of primary importance for possible glacial inception. The Cretaceous Antarctic palaeotopography still remains loosely constrained and is not the main focus of the reconstructions of ref. 36. Of most importance is essentially whether sufficiently high elevation was already in place over Antarctica at the time, particularly in the locations of the Gamburtsev and the Transantarctic Mountains. There is evidence that both mountain ranges already existed during the Cretaceous[54,55]. Three main uplift episodes have been found for

the Transantarctic Mountains between the Late Jurassic/Early Cretaceous and the early Eocene[54,56] while uplift phases for the Gamburtsev Mountains include the Permian and the mid-Cretaceous[55,57]. The magnitude of these uplift events is very complex to infer and has hence remained speculative, but recent evidence from ODP core 188-1166A suggests already elevated Gamburtsev Mountains during the Cretaceous[58]. To remain the most conservative in our approach and to ensure that our results are not related to local effects arising from different topographic reconstructions for each stage, we have chosen to patch the same Antarctic topography over the three reconstructions, while the bathymetry and topographic features located outside Antarctica were left untouched. We use the Antarctic topography provided by Wilson et al.[59] for the EO because this stage yet remains the oldest for which a realistic full topography is available for Antarctica.

We impose the same boundary conditions except the time-dependent solar constant, which is calculated by reducing the modern by 1% per 100 million years[37]. This yields solar constant values of 1,351.4 W m$^{-2}$ for the Aptian, 1,354 W m$^{-2}$ for the Cenomanian–Turonian and 1,357.4 W m$^{-2}$ for the Maastrichtian.

**Coupled climate-ice sheet methodology.** Several recent studies have shown that complex coupling methods between general circulation models (GCMs) and ice sheet models (ISMs) were required to provide relevant estimates of the ice volume, which can be accommodated over the high latitudes[26,60]. In particular, a new method has recently been proposed, which is designed for the study of transient glaciations and takes into account the albedo and height–mass balance feedbacks in addition to variations of the Earth's orbit and of atmospheric $CO_2$ levels[26], thereby representing a significant step forward in coupled climate-ice sheet modelling. Its application to the EO transition has given results well correlated with proxy data, in terms of $CO_2$ threshold for the glaciation and of temporal evolution of the ice sheet. This method, however, needs relatively well-documented variations of atmospheric $CO_2$ levels and orbital parameters and unfortunately suffers from exceedingly expensive computational costs.

Several issues presently preclude the use of such complex methods on the Cretaceous. First, due to the chaotic nature of the solar system[61], accurate orbital reconstructions do not exist for the Mesozoic. Second, atmospheric $CO_2$ levels are affected by large uncertainties and remain poorly resolved temporally[40].

In this study, we follow a different approach. Our purpose is to compare the sensitivity of three Cretaceous palaeogeographies to glacial onset and to provide estimates of the $CO_2$ levels required to trigger hundreds of thousands of years to million year-long stable ice sheets. We have applied a relatively simple one-way coupling method, which allows us to test the broad range of boundary conditions we dispose of, and which is explained with more details below. Moreover, it should be noted that another recent study has attested of the relevance of simple one-way coupling between climate and ice sheets regarding the estimation of atmospheric $CO_2$ thresholds for stable glacial onset[34] (see Data Repository, Section C), although simple one-way methods preclude any accurate estimation of accumulated ice volume.

**Models and method details.** The different climatic states used to force the ISM are obtained following ref. 26 and as follows: we use the Institut Pierre-Simon Laplace (IPSL) Atmospheric GCM LMDz[62], with a 3.75° × 1.9° resolution and 39 vertical levels, to generate temperature and precipitation fields. To incorporate the dynamics of the ocean, and because the time required to simulate three palaeogeographic time slices at different $CO_2$ levels with the high-resolution fully coupled IPSL-CM5 model is prohibitive, we use the mixed resolution (ocean: 2.8° × 1.4° and 24 vertical levels; atmosphere: 7.5° × 4.5° and 18 vertical levels) fully coupled Fast Ocean Atmosphere Model (FOAM)[63] as a sea-surface temperature (SST) generator. For each combination of palaeogeography and $CO_2$, we integrate the FOAM GCM for 2,000 years without deep ocean acceleration or flux corrections. During the last 100 years of each simulation, there is no apparent drift in the upper ocean and <0.05 °C per century change in globally averaged ocean temperature. Then, we average the SST field over the last 100 years and use it to force the higher-resolution AGCM LMDz for 20 years with identical boundary conditions. In both GCMs, orbital parameters are maintained constant to a favourable configuration for the inception of a glaciation in the Southern Hemisphere (eccentricity = 0.05, obliquity = 24.5° and perihelion in July). Temperature and precipitation fields are subsequently averaged over the last 5 years of simulation and used to drive the ISM GRISLI[64], which runs on a 40 km × 40 km grid. In the present work, the ice sheet model is run for 10 kyr keeping the temperature and precipitation forcings constant to evaluate potential ice sheet onset for each combination of palaeogeography and $CO_2$. The integration time of the ISM is chosen by approximating that, over a full precession cycle, favourable conditions for ice accumulation over the high latitudes occur during about half a cycle (∼10 kyr) and unfavourable conditions during the other half.

**Calibration of the one-way method.** The simple method described above does not include ice sheet feedbacks on climate. It is thus necessary to calibrate its results versus those deduced from more elaborated methods, which include these feedbacks. Because we recently applied such a method to the EO glaciation[26], we can easily compare the results obtained with the simple method with those obtained with the complex one. We thus carry out calibrating experiments using the same

procedure as described above. We use the EO palaeogeography from ref. 26, the same Southern Hemisphere cold summers orbital conditions and an adapted solar constant, while other boundary conditions are kept identical. We test four different atmospheric $CO_2$ levels: 1,120, 980, 840 and 560 p.p.m. These levels bracket the $CO_2$ threshold for a perennial Antarctic glaciation derived with our more complex method. This value of ∼925 p.p.m. is close to the $CO_2$ threshold obtained with other models and is also in agreement with palaeo-$CO_2$ data reconstructions (ref. 26, and reference therein). Note that, by perennial, we refer to a stable ice sheet lasting at least a few hundreds of thousands of years.

For each of these $CO_2$ levels, we run the GRISLI model for 10 kyr. We thus obtain four different ice sheet sizes, corresponding to 10 kyr of constant atmospheric forcing (Supplementary Fig. 4). The simple method thus gives ice sheet sizes of 0.24 and 1.44 millions of km$^2$ for 980 and 840 p.p.m., respectively, after 10 kyrs of constant forcing. Because the $CO_2$ threshold for a perennial ice sheet evaluated with the complex method is circa 925 p.p.m., this means that the ice sheet size of the EO 840 p.p.m. scenario (obtained with the simple method) is larger than what would be the minimal size required for the ice sheet to survive the changing orbital conditions and become perennial (while obviously, the EO 980 p.p.m. ice sheet size is lower).

This calibrating experiment can thus be seen as an indicator of the required minimal ice sheet size whose feedbacks will significantly impact the climate and lead to a perennial ice sheet. By subsequently comparing the ice sheet sizes obtained with the Cretaceous palaeogeographies with those obtained for the EO transition, we can deduce $CO_2$ threshold for perennial glaciation for each Cretaceous palaeogeography. For simplicity, and because the exact minimal ice sheet size that will remain perennial cannot simply be extrapolated from the EO 980 and 840 p.p.m. sizes, we assume here that to trigger the onset of a perennial ice sheet, the size obtained with the Cretaceous palaeogeographies must be larger or equal to the 840 p.p.m. size obtained with the EO palaeogeography (Supplementary Fig. 4). To summarize, although the ice sheets obtained after 10 kyr of ice sheet model experiment are certainly not in equilibrium with climate, if, after these 10 kyr, the ice sheet size is larger than the EO 840 p.p.m. ice sheet, we assume that its size will be large enough to stay perennial and survive changing orbital conditions.

As the ice sheets are not equilibrated after 10 kyr, the absolute values of ice sheet size and volume obtained for the different $CO_2$ values tested should not be used as reliable estimates. As stated above, reliable estimates would necessitate complex climate-ice sheet coupling forced by well-documented $CO_2$ and orbital records that are still lacking for the Cretaceous.

**Consistency with data and between models.** The main objective of the present study is to evaluate the sensitivity of different Cretaceous stages to ice sheet inception under various atmospheric $CO_2$ levels. As such, a direct and extensive comparison between our results and available data is beyond the scope of this study. However, it is still important and useful to determine whether our models can reasonably reproduce the mean climatic states observed in Cretaceous data. The FOAM model has recently been used to investigate how ocean circulation changes have impacted occurrences of Ocean Anoxic Events during the Cretaceous[44]. In this paper is presented an extensive comparison between the ocean temperatures modelled by FOAM and those inferred from proxy data. A good agreement is found (their Supplementary Fig. 7) demonstrating that the FOAM model reasonably simulates Cretaceous climates[65].

Unfortunately, the coarse atmospheric resolution of FOAM precludes direct use of its temperature and precipitation fields to force the ISM GRISLI and necessitates a higher-resolution model, here LMDz. It is not straightforward that the results obtained with the AGCM LMDz forced by SST fields generated with FOAM are consistent with the direct atmospheric outputs of FOAM. In the following is detailed a comparison between LMDz and FOAM atmospheric outputs, showing that the feedback analysis proposed in the main text is robust. Supplementary Fig. 11 displays the zonally averaged austral summer temperature difference between the Turonian and the Aptian at 560 p.p.m. in LMDz and in FOAM. Relative and specific humidity changes are also shown (in fact, the LMDz subplot is identical to the Fig. 3a of the main text). The difference between the Turonian and the Maastrichtian is shown on Supplementary Fig. 12, also in LMDz and FOAM. The Supplementary Fig. 13 displays the cloud changes in FOAM during the Turonian with respect to the Aptian or Maastrichtian. They are very similar to the cloud changes occurring in LMDz (Supplementary Fig. 9). These figures demonstrate that the Antarctic warming mechanism thwarting the development of ice sheets during the Turonian is also observed in FOAM. Finally, note that throughout the main text, atmospheric analyses are conducted with the LMDz model while oceanic analyses rely on the FOAM model.

**Code availability.** The codes of the FOAM, LMDz and GRISLI models are available on request to the authors. Note that the LMDz code can also be found at http://lmdz.lmd.jussieu.fr/.

**Data availability.** Data that support the results of this study are available on request to the authors.

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

## Acknowledgements

We are indebted to Christophe Dumas for assistance with the ice sheet model. Pierre Sepulchre, Alexandre Pohl, Didier Paillard, Masa Kageyama and Guillaume Dupont-Nivet are acknowledged for discussions and advice. We thank the two anonymous reviewers for the quality of their comments that greatly improved the manuscript. We thank the CEA/CCRT for providing access to the HPC resources of TGCC under the allocation 2014-012212 made by GENCI. We acknowledge support from the Anox-Sea project funded by the ANR under grant ANR-12-BS06-0011-03.

## Author contributions

J.-B.L. designed and performed the numerical simulations; Y.D. conceived the project. Both authors analysed and discussed the results, and wrote the manuscript.

## Additional information

**Competing financial interests:** The authors declare no competing financial interests.

