## [Peer review file · Nature Communications]

Reviewers' comments:

Reviewer #1 (Remarks to the Author):

In this paper, Ladant and Donnadiou show that changing paleogeography during the Cretaceous can control the size of an Antarctic ice sheet. The paper presents some new and interesting results which I think could eventually be suitable for publication in Nature Communications. The methods are appropriate and have been robustly presented and evaluated in previous studies. However, I do have some important comments and questions.

Main comments

At the moment I am not totally convinced that the threshold for glaciation is hugely different from the Turonian compared to the other stages. It could be that the CO₂ threshold for glaciation in the Turonian is just below 560ppm (e.g. 559ppmv!), and the threshold for the Aptian is just above 560 ppmv (e.g. 561 ppmv!). I would be much happier with the conclusions of the paper if there were (several) more CO₂ levels tested in the range e.g. 400 ppmv to 840 ppmv.

Line 87 - the assumption that temperature is the dominant effect could be tested by running ice sheet simulations with e.g. the temperature of the Turonian with the precipitation of the Aptian.

Most climate models show very high interannual variability over Antarctica. I would like the authors to demonstrate that the temperature differences between the different Stages over Antarctica are statistically significantly different to each other, and that the results of the ice sheet model still hold within the uncertainty due to interannual variability.

The main result is that the Turonian has a lower CO₂ threshold for glaciation than the other Stages. This is due to the different paleogeographies. However, how confident are we of these paleogeographies, and in particular, how confident are we in the very subtle differences which give rise to the different CO₂ thresholds? I suspect that the answer is 'not very'.

Specific Comments

Line 18. I do not agree with this. The work has little or no relation to the future in my opinion - e.g. there is insufficient data to constrain the results to allow validation of the ice sheet model. At the end of the abstract I think two more things are needed: (1) something about the comparison with data, and (2) A final statement stating that therefore paleogeography can have a strong effect on global climate by controlling the threshold for ice sheet inception.

Figure S3 - need to clarify in the text and caption if this is the mass balance at the end of the ice sheet simulation or at the beginning. If at the end, the differences are exaggerated because they include the effect of the ice itself on mass balance via the lapse-rate effect.

Line 99-101 - some explanation should be given of why 'fragmented continents' leads to a warming.

Line 115 - The changes in ocean circulation need to be directly linked back to the differences in paleogeographies in the different Stages.

Line 184-189. I don't agree with this. The temperature and CO₂ records of this time are both too sparse and have large uncertainties - as such I do not think you can say that paleogeography is the main driver rather than CO₂, nor that any temperature trend is 'confirmed'.

Line 271-273. The orbits chosen are not representative of an average over 10kyr of a precession cycle. I expect instead they represent close to the maximum - as such I don't think you can use the argument that the time length is 'roughly the time during which' the applied orbits would have lasted.

Technical comments and typos

Line 13-15. Polar distributions of ****what is today**** low latitude fauna and flora.

Line 15-18. This sentence does not make sense, and I am not sure why there is a 'yet' in the sentence.

Line 21 - instead of 'modest' which could mean anything, give a number. Also I do not understand why this is 'in spite of'.

Line 30 - 'several degrees' - please be more precise - give a range.

Line 48 - 'encompassing values' rather than 'comparable to those'.

Line 142 - not sure of the use of 'concur' here.

Line 153 - 2002 is not recent!

Line 164 - not sure what 'makes no consensus yet' means

Reviewer #2 (Remarks to the Author):

This paper explores the role of paleogeography in regulating past glaciations during the Cretaceous using coupled climate models and an ice sheet model. In particular a decoupling between atmospheric CO₂ forcing of glaciations is identified due to complex ocean-atmosphere feedbacks (similar to that proposed by Rose and Ferreira, 2013) for certain paleogeographic configurations. The role on non-CO₂ forcing of paleoclimate is an important and interesting point. There is a thorough discussion of the mechanisms that lead to much warmer conditions at high latitudes for certain paleogeographies. The results also lend support to the hypothesis that ice sheets may have existed during certain stages (the late Aptian and Maastrichtian) although not others (the Turonian) of the Cretaceous. However, the uncertainties associated with Cretaceous CO₂ reconstructions and data supporting Cretaceous glaciation are not adequately discussed. The paper is generally well written, although certain sentences are hard to understand (see minor points). The methodology, use of statistics, and presentation of results are sound.

Main points (validity of conclusions and suggested improvements):

The authors suggest that paleogeographic changes lead to a much lower Antarctic glacial CO₂ threshold for the Turonian, compared with the Aptian and Maastrichtian. Mechanisms similar to the conceptual model of Rose and Ferreira (2013) are identified as the cause of much warmer high latitudes for the Turonian. This discussion of this is very good and detailed, however there is no mention of why the relatively small changes in paleogeography from the Aptian to Turonian to Maastrichtian may generate this response. Although this may not be understood, the paper would be more rewarding to read if there was some discussion or speculation as to why the Turonian paleogeography generates this response. Otherwise it is hard not to view this as a potentially model dependent result.

The potential for Cretaceous Antarctic ice sheets is discussed, with the results supporting ice sheets during the Aptian and Maastrichtian but not the Turonian. One conclusion (line 180) is that sequence stratigraphy and oxygen isotope excursions may not be reliable indicators of glacio-eustasy due to the disagreement with the model results. The reliability of these records as indicators of glacio-eustasy has been discussed at length elsewhere and there should be some discussion or at least reference to some of the reasons why these records may not be reliable indicators of Cretaceous glacio-eustasy.

As discussed, whether there were ice sheets or not in the Cretaceous is also dependent on absolute atmospheric CO₂ concentrations, which are poorly constrained. Although there is some discussion of Cretaceous CO₂ uncertainty, in particular in determining absolute values, this could be more detailed. However, although absolute values are hard to determine, the main result of the paper (that the glacial CO₂ threshold may differ due to changes in paleogeography) may still be robust because it is not dependent on absolute values. I would therefore suggest placing less

emphasis on absolute CO₂ values and more on the relative difference in CO₂ thresholds between the Aptian/Maastrichtian and Turonian.

In the abstract it is stated that 'this issue is becoming critical regarding forecasts of a mostly ice-free future', suggesting that the Cretaceous is an analog for future warming. I would suggest removing this statement as it is not discussed in the main paper and the main conclusion of the paper highlights that the Cretaceous probably isn't a suitable analog.

Minor points:

Geological stages: include definitions of geological stages and check age of Turonian.

15-17: Hard to understand this sentence, suggest: However, recent data hint at the possibility of glacial events, although questions remain as to how perennial ice can accumulate at the poles under warm climates of the Cretaceous.

18: "This issue is becoming critical..." suggest removing this sentence unless this point is made and backed up in the main paper.

21: "Modest atmospheric CO₂", this is quite subjective, and 560 ppm may not be considered modest in a lot of fields. Suggest, 'atmospheric CO₂ well below the Antarctic glacial threshold suggested by other studies'?

45: "and may benefit from additional support provided by the revision of atmospheric pCO₂ estimate during the Cretaceous", not very clear, suggest mentioning that pCO₂ estimates have been revised downwards in ref. 22. Also check that ref. 22 is relevant to the Turonian, as the sentence currently suggests.

48: define Eocene-Oligocene transition.

64: "a well-constrained methodology", clarify what is meant by this.

73: From the current simulations and figures it is not clear at what point the Turonian paleogeography would glaciates, all that can be said is that Antarctic glaciation is below 560 ppm. If you could calculate an approximate glacial threshold, or include a simulation at lower CO₂ for the Turonian it would be a stronger argument. Looking at the existing figures I expect that the Turonian glacial threshold is well below 560 ppm.

164: 'makes no consensus yet', not clear what is meant. Suggest 'On the other hand, there is no consensus yet in coupled modeling studies regarding the Turonian to Maastrichtian cooling seen in the data'?

180-181: Need discussion of problems in interpreting regional sea level records and oxygen isotopes as a glacio-eustatic signal.

210: Suggest rewording 'is barely discussed'

215: Although I agree that this is probably the most reasonable approach, I would suggest including some discussion of the ages of the mountain ranges where ice first accumulates in your simulations.

238: This is a problem regardless of the methodology used and should be discussed in more depth in the main paper.

273: Although the ice sheet integration time of half a precession cycle (10 kyr) may be a fair

assumption, it should be made clear that the ice sheets shown are far from equilibrium, as is shown from Figure S8. Also slightly confusing with the definition of 'perennial ice' as a stable ice sheet that lasts at least a few 100 kyrs old, something that isn't tested.

Point by point response to the reviews of manuscript “Palaeogeographic regulation of glacial events during the Cretaceous Supergreenhouse”

Reviewer #1 (Remarks to the Author)

In this paper, Ladant and Donnadieu show that changing paleogeography during the Cretaceous can control the size of an Antarctic ice sheet. The paper presents some new and interesting results which I think could eventually be suitable for publication in Nature Communications. The methods are appropriate and have been robustly presented and evaluated in previous studies. However, I do have some important comments and questions.

Thank you for this comment.

Main comments

At the moment I am not totally convinced that the threshold for glaciation is hugely different from the Turonian compared to the other stages. It could be that the CO₂ threshold for glaciation in the Turonian is just below 560ppm (e.g. 559ppmv!), and the threshold for the Aptian is just above 560 ppmv (e.g. 561 ppmv!). I would be much happier with the conclusions of the paper if there were (several) more CO₂ levels tested in the range e.g. 400 ppmv to 840 ppmv.

We agree with the reviewer that our manuscript will be stronger with additional experiments at other CO₂ levels to refine our glacial threshold values.

For the three palaeogeographies, we tested two other CO₂ levels: 650 ppm (~ 2.3 PAL) and 750 ppm (~ 2.7 ppm). In addition, but only for the Turonian configuration, we tested CO₂ levels of 420 ppm (1.5 PAL) and 280 ppm to investigate its threshold for glaciation.

Results from these additional experiments demonstrate that the threshold for glaciation is comprised between 750 and 840 ppm for the Aptian, between 650 and 750 ppm for the Maastrichtian and between 280 and 420 ppm for the Turonian, thereby differences in threshold of at least 350 ppm between the Aptian and the Turonian (Figs. S2 and S10 of the revised Supplementary Materials). These additional simulations thus strengthen our main conclusions. See also lines 81-93 of the revised manuscript.

Line 87 - the assumption that temperature is the dominant effect could be tested by running ice sheet simulations with e.g. the temperature of the Turonian with the precipitation of the Aptian.

Two sensitivity experiments with a Turonian palaeogeography have been carried out to confirm this assumption, even though we think that our way of demonstrating it in the original manuscript, via the ablation term (Fig. S3 of the original and Fig. S4 of the revised manuscript respectively), remains meaningful.

In the first (second) experiment, we prescribed the temperature field of the 560 ppm Turonian (Aptian) simulation and the precipitation field of the Aptian (Turonian) simulation. As expected, there is no accumulation of ice in the simulation with the Turonian temperature

fields, contrary to the other (Fig. 1 of this response). Please also refer to lines 101-106 of the revised manuscript and Fig. S5 of the revised Supplementary Materials.

Most climate models show very high interannual variability over Antarctica. I would like the authors to demonstrate that the temperature differences between the different Stages over Antarctica are statistically significantly different to each other, and that the results of the ice sheet model still hold within the uncertainty due to interannual variability.

For each CO₂ levels, we have performed Student t-tests to check whether the temperature differences over Antarctica between the Stages are statistically significant with respect to the interannual variability. The Figures 2 and 3 of this response show the statistically significant (at 95% confidence) SST and temperature at 2 m (T2M) differences between the Turonian and the Aptian and the Turonian and the Maastrichtian for the 560 ppm simulations. The SST differences are computed with FOAM and the T2M differences with LMDz. This confirms that the temperature differences between the Stages are significant over the whole Antarctica. We hence have changed the Figure 2 of the revised manuscript, which now displays the statistically significant T2M differences.

The main result is that the Turonian has a lower CO₂ threshold for glaciation than the other Stages. This is due to the different palaeogeographies. However, how confident are we of these palaeogeographies, and in particular, how confident are we in the very subtle differences which give rise to the different CO₂ thresholds? I suspect that the answer is 'not very'.

We totally agree with the fact that palaeogeographies are subject to uncertainties. We used the palaeogeographies from the Sewall et al. (2007) paper as these are the most up to date that are currently available in our group (we are aware that there are alternate palaeogeographies of the Cretaceous used in the literature, e.g., Lunt et al. (2016), but these are proprietary and thus not freely available to us).

We are confident about our main result however, as the ocean-atmosphere feedbacks that are responsible for the Turonian warmth originate from palaeogeographic differences that seem robust across the literature.

Indeed, the Aptian to Turonian Antarctic warming is primarily caused by the opening of the equatorial Atlantic gateway following the separation between South America and Africa, as also demonstrated by Poulsen et al. (2003). The closed gateway between North and South Atlantic in the Aptian and its open state in the Turonian is a consistent feature across palaeogeographic reconstructions (e.g., the Cretaceous maps of Lunt et al. 2016).

The Turonian to Maastrichtian Antarctic cooling is also attributed to ocean circulation changes. Those have been extensively investigated in Donnadieu et al. (2016), and have been evaluated against different gateways configurations, thus bringing additional confidence to our main conclusions.

Specific Comments

Line 18. I do not agree with this. The work has little or no relation to the future in my opinion - e.g. there is insufficient data to constrain the results to allow validation of the ice sheet model.

We agree and have removed this sentence.

At the end of the abstract I think two more things are needed: (1) something about the comparison with data, and (2) A final statement stating that therefore paleogeography can have a strong effect on global climate by controlling the threshold for ice sheet inception.

The revised abstract has been modified and shortened to comply with the editorial policy of *Nature Communications*. These two points are addressed lines 22-25 of the revised manuscript.

Figure S3 - need to clarify in the text and caption if this is the mass balance at the end of the ice sheet simulation or at the beginning. If at the end, the differences are exaggerated because they include the effect of the ice itself on mass balance via the lapse-rate effect.

The mass balance, accumulation and ablation terms are calculated at the start of the IS simulation (caption of Fig. S3 was: “Antarctic initial mass balance, ...”).

We thank you however for pointing out that it was not very clear so we changed the caption to (see Fig. S4 of the revised Supplementary Materials):

“Antarctic mass balance, accumulation and ablation terms for each palaeogeography at the beginning of the ice sheet simulations.”

Line 99-101 - some explanation should be given of why 'fragmented continents' leads to a warming.

OK. We have added a short explanation (lines 116-121 of the revised manuscript).

Line 115 - The changes in ocean circulation need to be directly linked back to the differences in paleogeographies in the different Stages.

Thank you for pointing this out. We indeed unintentionally omitted this step in the initial manuscript.

The Aptian to Turonian South Atlantic and Indian Ocean warming is attributed to the opening of the equatorial Atlantic seaway, as demonstrated by Poulsen et al. (2003), who showed that the opening of this gateway led to the export of warm and saline upper ocean waters into the South Atlantic, triggering deep water formation in the South Atlantic and a strong ocean warming, thereby increasing the extratropical heat transport.

The Turonian to Maastrichtian ocean circulation changes have been studied in details in the recent study of Donnadieu et al. (2016). In this work, the authors argue that palaeogeographic changes between the Turonian and the Maastrichtian favour the intensification of deep-water production in the South Atlantic and Indian Ocean by means of changes in the South Atlantic basin hydrological cycle and of modifications in the configuration of the Caribbean Seaway between the Turonian and the Maastrichtian, these latter triggering the development of a strong westward water flow at all depth through the Seaway.

We thus have modified the manuscript on lines 129-143 of the revised manuscript.

Line 184-189. I don't agree with this. The temperature and CO₂ records of this time are both too sparse and have large uncertainties - as such I do not think you can say that paleogeography is the main driver rather than CO₂, nor that any temperature trend is 'confirmed'.

We agree that both temperature and CO₂ records are sparse and display large uncertainties but it seems robust between the studies that have investigated the long-term temperature and CO₂ variations that the Turonian has warmer temperature and CO₂ values than the Aptian and the Maastrichtian (e.g., Clarke and Jenkyns 1999, Ditchfield et al. 1994, Pucéat et al. 2003, Breecker et al. 2010, Wang et al. 2014). Whether temperature and/or CO₂ records have been calibrated, within their own large uncertainties, to reproduce the trend from older temperature and/or CO₂ records is certainly a critical issue, which may affect conclusions of our and other studies, but it is not the purpose of this work to investigate it.

Still, it is interesting to note that the mean global annual temperature from the Aptian to the Turonian to the Maastrichtian presents a bell-like evolution with highest temperatures in the Turonian, which resembles roughly the temperature trend extracted from data studies. Speaking about a “confirmation” of the temperature trend is indeed presumptuous and we recognise that, considering the uncertainties, we are unable to confirm anything. We still note that the trend from our models seems in agreement with data without invoking any CO₂ variation.

Finally, it is true that we cannot conclude whether palaeogeography is the main driver of this trend or not nor whether CO₂ is. As such, we hope that the re-phrasing of our manuscript will not give this impression upon reading (see, e.g., lines 202-208 of the revised manuscript).

Line 271-273. The orbits chosen are not representative of an average over 10kyr of a precession cycle. I expect instead they represent close to the maximum - as such I don't think you can use the argument that the time length is 'roughly the time during which' the applied orbits would have lasted.

Although the orbit we prescribed indeed creates favourable conditions for ice accumulation, we have not chosen the most extreme orbit in terms of mean summer insolation (illustrated for instance over the arbitrarily defined interval 42.5 – 37.5 Ma, Fig. 4 of this response). In addition, we do not state that the time length is ‘roughly the time during which the applied orbits would have lasted’ but ‘roughly the time during which favourable conditions for ice accumulation would occur’, which is quite different. Considering a precession cycle of 20 kyrs, the conditions for ice sheet growth can be crudely approximated to relatively favourable during half a cycle (10 kyrs) and relatively unfavourable during the other half.

Thus, even if not completely physically correct, we made the approximation that during extreme insolation minima, the time length during which ice would accumulate over a precession cycle (even if not at a constant rate) could be represented by a 10 kyrs time length under our constant orbital parameters.

We have tried to clarify it in the revised manuscript by rephrasing the sentence on the integration time of the ISM (see Methods, section Models and method details, lines 304-326).

Technical comments and typos

Line 13-15. Polar distributions of ****what is today**** low latitude fauna and flora.

The abstract has been rephrased.

Line 15-18. This sentence does not make sense, and I am not sure why there is a 'yet' in the sentence.

This sentence has been reformulated (see abstract of the revised manuscript).

Line 21 - instead of 'modest' which could mean anything, give a number. Also I do not understand why this is 'in spite of'.

This sentence has been reformulated (see abstract of the revised manuscript).

Line 30 - 'several degrees' - please be more precise - give a range.

Done (lines 28-29 of the revised manuscript).

Line 48 - 'encompassing values' rather than 'comparable to those'.

Corrected (line 46-47 of the revised manuscript).

Line 142 - not sure of the use of 'concur' here.

This has been changed to “add up” (line 171 of the revised manuscript).

Line 153 - 2002 is not recent!

Yes, that is true. “Recent” has been removed.

Line 164 - not sure what 'makes no consensus yet' means

The sentence has been reformulated (line 193-194 of the revised manuscript).

Reviewer #2 (Remarks to the Author):

This paper explores the role of paleogeography in regulating past glaciations during the Cretaceous using coupled climate models and an ice sheet model. In particular a decoupling between atmospheric CO₂ forcing of glaciations is identified due to complex ocean-atmosphere feedbacks (similar to that proposed by Rose and Ferreira, 2013) for certain paleogeographic configurations. The role on non-CO₂ forcing of paleoclimate is an important and interesting point. There is a thorough discussion of the mechanisms that lead to much warmer conditions at high latitudes for certain paleogeographies. The results also lend support to the hypothesis that ice sheets may have existed during certain stages (the late Aptian and Maastrichtian) although not others (the Turonian) of the Cretaceous. However, the uncertainties associated with Cretaceous CO₂ reconstructions and data supporting Cretaceous glaciation are not adequately discussed. The paper is generally well written, although certain sentences are hard to understand (see minor points). The methodology, use of statistics, and presentation of results are sound.

Thank you for this comment.

Main points (validity of conclusions and suggested improvements):

The authors suggest that paleogeographic changes lead to a much lower Antarctic glacial CO₂ threshold for the Turonian, compared with the Aptian and Maastrichtian. Mechanisms similar

to the conceptual model of Rose and Ferreira (2013) are identified as the cause of much warmer high latitudes for the Turonian. This discussion of this is very good and detailed, however there is no mention of why the relatively small changes in paleogeography from the Aptian to Turonian to Maastrichtian may generate this response. Although this may not be understood, the paper would be more rewarding to read if there was some discussion or speculation as to why the Turonian paleogeography generates this response. Otherwise it is hard not to view this as a potentially model dependent result.

We thank the reviewer for this comment and for pointing out this missing link. This is what we have added in the revised manuscript (lines 129-143):

“These ocean changes are closely correlated to the onset or shutdown of convective mixing areas – interpreted as deep-water formation zones – in the Southern Ocean (Fig. 2 and Fig. S6). Deep-water formation increases advection of warm low-latitude surface waters, the radiative cooling of which (especially during winter) generates vertical mixing between these waters and the warmer subsurface. This efficiently prevents sea-ice formation and limits the cooling of the ocean at high-latitude³⁹, resulting in strong warming anomalies. These differences in convective mixing zones are directly linked to the differences in palaeogeographies. First, the opening of the equatorial Atlantic gateway between Africa and South America in the transition from the Aptian to the Turonian generates an export of warm and saline upper ocean waters into the South Atlantic, triggering deep-water formation and a strong ocean warming (see ref. 40). Second, changes in ocean circulation and areas of deep-water formation between the Turonian and the Maastrichtian have recently been investigated and attributed to modifications of the South Atlantic hydrological cycle and of the Caribbean Seaway configuration⁴¹”.

The potential for Cretaceous Antarctic ice sheets is discussed, with the results supporting ice sheets during the Aptian and Maastrichtian but not the Turonian. One conclusion (line 180) is that sequence stratigraphy and oxygen isotope excursions may not be reliable indicators of glacio-eustasy due to the disagreement with the model results. The reliability of these records as indicators of glacio-eustasy has been discussed at length elsewhere and there should be some discussion or at least reference to some of the reasons why these records may not be reliable indicators of Cretaceous glacio-eustasy.

We agree with reviewer 2 on this but want to clarify that we do not challenge the reliability *in general* of sequence stratigraphy and oxygen isotopes excursions in recording past ice sheets evidence. Over the Turonian Stage however, in the absence of direct evidence of glaciations, there are intense debates in the data community (e.g., Miller 2009) as some records find evidence for ice sheets (e.g., Bornemann et al. 2008, Galeotti et al. 2009) whereas others do not (e.g., Moriya et al. 2007, Ando et al. 2009, MacLeod et al. 2013).

We acknowledge that we have been presumptuous in the first version of our manuscript because our findings do not unquestionably demonstrate that ice sheets were not present during the Cenomanian-Turonian. It was thus rather unfair to question the reliability of previous studies arguing in favour of Cenomanian-Turonian ice sheets. In the revised version, we now stipulate that our results better corroborate previous work arguing against Cenomanian-Turonian ice sheets but that new data studies are needed to resolve this debate (lines 212-228 of the revised manuscript).

We finally note that arguments as to what (other than ice sheets) could be recorded by data studies in favour of Cenomanian-Turonian ice sheets are given by MacLeod and colleagues (2013), to which we make extensive reference in our manuscript. Entering into the debate

about the reliability of proxy records is not our purpose here, but should you think that more references are needed, we would be happy to incorporate those you believe important.

As discussed, whether there were ice sheets or not in the Cretaceous is also dependent on absolute atmospheric CO₂ concentrations, which are poorly constrained. Although there is some discussion of Cretaceous CO₂ uncertainty, in particular in determining absolute values, this could be more detailed. However, although absolute values are hard to determine, the main result of the paper (that the glacial CO₂ threshold may differ due to changes in paleogeography) may still be robust because it is not dependent on absolute values. I would therefore suggest placing less emphasis on absolute CO₂ values and more on the relative difference in CO₂ thresholds between the Aptian/Maastrichtian and Turonian.

We agree and hope that the rephrasing throughout the revised manuscript will adequately answer this comment.

In the abstract it is stated that 'this issue is becoming critical regarding forecasts of a mostly ice-free future', suggesting that the Cretaceous is an analog for future warming. I would suggest removing this statement as it is not discussed in the main paper and the main conclusion of the paper highlights that the Cretaceous probably isn't a suitable analog.

Yes, this sentence has been removed.

Minor points:

Geological stages: include definitions of geological stages and check age of Turonian.

We thank the reviewer for pointing this out. Indeed, the date of 95 Ma actually falls at the very end of the Cenomanian Stage. However, considering that 1) palaeogeographies for the Cretaceous are still uncertain and 2) at the model resolution it is not really possible to differentiate palaeogeographies separated by only a few million years (except opening/closing of gateways), our late Cenomanian palaeogeography can probably fairly well represent a Turonian palaeogeography as well as a mid-Cenomanian palaeogeography.

In the original text, we actually first referred to the Cenomanian-Turonian palaeogeography (line 58 of the original version of the manuscript) before simply referring to the Turonian, essentially because it is the stage that has been the main focus of a debate about the existence of ice sheets and that is considered the Cretaceous Climatic Optimum.

In the revised version, we have tried to be more careful and to speak about the Cenomanian-Turonian Stage whenever necessary (i.e., in the Introduction, see lines 27-48 of the revised manuscript, and the second part of the Discussion/Conclusion, see lines 209-242 of the revised manuscript).

15-17: Hard to understand this sentence, suggest: However, recent data hint at the possibility of glacial events, although questions remain as to how perennial ice can accumulate at the poles under warm climates of the Cretaceous.

Thank you for this suggestion, which has been implemented and modified (see revised abstract). Please note that the revised abstract has also been shortened to comply with the editorial policy of *Nature Communications*.

18: "This issue is becoming critical..." suggest removing this sentence unless this point is made and backed up in the main paper.

This has been removed.

21: "Modest atmospheric CO₂", this is quite subjective, and 560 ppm may not be considered modest in a lot of fields. Suggest, 'atmospheric CO₂ well below the Antarctic glacial threshold suggested by other studies'?

This is true, thanks. It has been corrected (see the revised abstract).

45: "and may benefit from additional support provided by the revision of atmospheric pCO₂ estimate during the Cretaceous", not very clear, suggest mentioning that pCO₂ estimates have been revised downwards in ref. 22. Also check that ref. 22 is relevant to the Turonian, as the sentence currently suggests.

This sentence has been rephrased. Many thanks for pointing out that the ref. 22 does indeed suggest a downward revision of CO₂ estimates during the Cretaceous but in their figure, there is no record of Turonian age. We have thus changed this reference to Fletcher et al. 2005, Fletcher et al. 2008 and Barclay et al. 2010, that suggest CO₂ estimates of 500 – 1400 ppm for the mid-Cretaceous (although with large uncertainties). Please refer to lines 43-48 of the revised manuscript.

48: define Eocene-Oligocene transition.

We have added the age of the transition but we are unsure as to how you would like us to define it. It is written in the manuscript that a major Antarctic glaciation occurred at this time but we do not think more is needed (lines 47-48 of the revised manuscript).

64: "a well-constrained methodology", clarify what is meant by this.

This sentence was modified to clarify it (lines 67-70 of the revised manuscript).

73: From the current simulations and figures it is not clear at what point the Turonian paleogeography would glaciolate, all that can be said is that Antarctic glaciation is below 560 ppm. If you could calculate an approximate glacial threshold, or include a simulation at lower CO₂ for the Turonian it would be a stronger argument. Looking at the existing figures I expect that the Turonian glacial threshold is well below 560 ppm.

We agree that our manuscript would be stronger with a simulation that presents a glacial Turonian. We have thus performed several new simulations to precise glacial thresholds (see also the response to reviewer 1). For the Turonian in particular, we have tested new CO₂ levels of 420 ppm (1.5 PAL) and 280 ppm. As you indeed inferred, the threshold for glacial inception in the Turonian occurs between 420 and 280 ppm, that is roughly 400 ppm lower than in the Aptian and 300 ppm lower than in the Maastrichtian (lines 81-89 of the revised manuscript and Figs. S2 and S10 of the revised Supplementary Material).

164: 'makes no consensus yet', not clear what is meant. Suggest 'On the other hand, there is no

consensus yet in coupled modeling studies regarding the Turonian to Maastrichtian cooling seen in the data"?

Thank you for this suggestion. It is exactly what we meant (lines 193-194 of the revised manuscript).

180-181: Need discussion of problems in interpreting regional sea level records and oxygen isotopes as a glacio-eustatic signal.

Yes. Please refer to our response to one of your main points above (lines 212-228 of the revised manuscript).

210: Suggest rewording 'is barely discussed'

Done (lines 254-255 of the revised manuscript).

215: Although I agree that this is probably the most reasonable approach, I would suggest including some discussion of the ages of the mountain ranges where ice first accumulates in your simulations.

This is a very good suggestion. Some estimates regarding the age of the mountain ranges, essentially the Transantarctic Mountains (TAM) and the Gamburtsev Mountains (GM), and their uplift have been proposed, yet are still affected by very large uncertainties.

An extensive and very complete review has recently been published regarding the geological and tectonic evolution of the TAM (Elliot 2013), providing arguably the state of the art of what is known today concerning this major mountain range. During the Mesozoic, a first exhumation episode may have occurred in the Late Jurassic/Early Cretaceous but this episode is still uncertain as only fission track dating provides evidence of it. A second, more certain and more widespread, has occurred in the mid-Cretaceous before a major uplift event in the early Eocene (Elliot 2013 and reference therein, see notably Fitzgerald 2002).

Prior to the AGAP (Antarctic Gamburtsev Province) project, the origin of the GM was loosely constrained with ages ranging from the Cambrian to the Cenozoic (Cox et al. 2010, Rose et al. 2013). Results from the AGAP project have led to new studies trying to refine these estimates (e.g., Ferraccioli et al. 2011, Rose et al. 2013). The GM, that cover much of the East Antarctic continent, are now thought to originate back to ~ 1 Ga before being largely eroded. New phases of uplift are thought to have notably occurred during the Permian and the mid-Cretaceous, possibly giving rise to about 2 km of uplift (Ferraccioli et al. 2011, their figure 4, Rose et al. 2013).

We have added a discussion in the Methods section (lines 255-269 of the revised manuscript).

238: This is a problem regardless of the methodology used and should be discussed in more depth in the main paper.

In the new manuscript, we indicate several times that the CO₂ levels during the Cretaceous are uncertain and thus should be taken with caution. It has been shown that the need for well resolved CO₂ variations is critical to be able to go beyond oneway climate-ice sheet experiments (Pollard 2010, Ladant et al. 2014) but we think that this issue is adequately discussed in the Method Section (lines 276-291 of the revised manuscript).

273: Although the ice sheet integration time of half a precession cycle (10 kyr) may be a fair assumption, it should be made clear that the ice sheets shown are far from equilibrium, as is shown from Figure S8. Also slightly confusing with the definition of 'perennial ice' as a stable ice sheet that lasts at least a few 100 kyrs old, something that isn't tested.

We agree that the ice sheets shown are not in equilibrium.

However, although not explicitly tested in our manuscript, the comparison between our simple method and the complex one presented in another paper (Ladant et al. 2014) allows us to give insights about perennial ice sheets. Indeed, by applying our complex method on the EO, we have shown that the CO₂ threshold for a stable (few 100 kyrs) ice sheet is about 925 ppm. By applying the simple method presented in this manuscript on the EO and for different CO₂, we obtained, after 10 kyrs of constant orbit, non-equilibrated ice sheet sizes for 840 ppm and 980 ppm of pCO₂. This means that the non-equilibrated ice sheet size of the 840 ppm scenario is larger than what would be the minimal non-equilibrated size required so that the ice sheet becomes perennial (while of course, the 980 ppm size is lower than the minimal size).

For simplicity here, we defined the EO 840 ppm ice sheet size to be the minimal that a (Cretaceous) ice sheet size needs to cross after 10 kyrs of integration to stay perennial. We could of course have run the ice sheet experiments longer but 1) the results would not have changed, as the forcing is constant and 2) we think it would have added confusion, because, if the orbital conditions for ice accumulation can approximately be favourable for 10 kyrs, this is certainly not the case for a longer integration.

We have rephrased the section about the calibration of the method to try to make it clearer, plus we have added a sentence at the end of the paragraph to summarise it (lines 343-368 of the revised manuscript).

We also have removed the paragraph about “perennial” vs “ephemeral” ice sheets. We now only speak about perennial ice sheets, with the meaning of ice sheets lasting at least a few 100 kyrs.

We also have added in the figure caption the fact that the ice sheets shown are not in equilibrium, as suggested (Fig. S10 of the revised Supplementary Materials).

References

- Ando, A. et al. Blake Nose stable isotopic evidence against the mid-Cenomanian glaciation hypothesis. *Geology* **37** (2009).
- Barclay, R. S. et al. Carbon sequestration activated by a volcanic CO₂ pulse during Ocean Anoxic Event 2. *Nature Geoscience* **3** (2010).
- Bornemann, A. et al. Isotopic evidence for glaciation during the Cretaceous supergreenhouse. *Science* **319** (2008).
- Breecker, D. O. et al. Atmospheric CO₂ concentrations during ancient greenhouse climates were similar to those predicted for 2100 A.D. *Proceedings of the National Academy of Science USA* **107** (2010).
- Clarke, L. J. & Jenkyns, H. C. New oxygen isotope evidence for long-term Cretaceous climatic change in the Southern Hemisphere. *Geology* **27** (1999).
- Cox, S. E. et al. Extremely low long-term erosion rates around the Gamburtsev Mountains in interior East Antarctica. *Geophysical Research Letters* **37** (2010).
- Ditchfield, P. W. et al. High latitude palaeotemperature variation: New data from the Thithonian to Eocene of James Ross Island, Antarctica. *Palaeogeography, Palaeoclimatology, Palaeoecology* **107** (1994).
- Donnadieu, Y. et al. A better-ventilated ocean triggered by Late Cretaceous changes in continental configuration. *Nature Communications* **7** (2016).
- Elliot, D. The geological and tectonic evolution of the Transantarctic Mountains: a review. *Geological Society, London, Special Publications* **381** (2013).
- Ferraccioli, F. et al. East Antarctic rifting triggers uplift of the Gamburtsev Mountains. *Nature* **479** (2011).
- Fitzgerald, P. Tectonics and landscape evolution of the Antarctic plate since the breakup of Gondwana, with an emphasis on the West Antarctic Rift System and the Transantarctic Mountains. *Royal Society of New Zealand Bulletin* **35** (2002).
- Fletcher, B. J. et al. Fossil bryophytes as recorders of ancient CO₂ levels: Experimental evidence and a Cretaceous case study. *Global Biogeochemical Cycles* **19** (2005).
- Fletcher, B. J. et al. Atmospheric carbon dioxide linked with Mesozoic and early Cenozoic climate change. *Nature Geoscience* **1** (2008).
- Galeotti, S. et al. Sea-level control on facies architecture in the Cenomanian–Coniacian Apulian margin (Western Tethys): A record of glacio-eustatic fluctuations during the Cretaceous greenhouse? *Palaeogeography, Palaeoclimatology, Palaeoecology* **276** (2009).
- Ladant, J.-B. et al. The respective role of atmospheric carbon dioxide and orbital parameters on ice sheet evolution at the Eocene-Oligocene transition. *Paleoceanography* **29** (2014).

- Laskar, J. et al. A long term numerical solution for the insolation quantities of the Earth. *Astronomy and Astrophysics* **428**, 261-185 (2004).
- Lunt, D. J. et al. Palaeogeographic controls on climate and proxy interpretation. *Climate of the Past* **12** (2016).
- MacLeod, K. G. et al. A stable and hot Turonian without glacial ^{18}O excursions is indicated by exquisitely preserved Tanzanian foraminifera. *Geology* **41** (2013).
- Miller, K. G. Palaeoceanography: Broken greenhouse windows. *Nature Geoscience* **2** (2009).
- Moriya, K. et al. Testing for ice sheets during the mid-Cretaceous greenhouse using glassy foraminiferal calcite from the mid-Cenomanian tropics on Demerara Rise. *Geology* **35** (2007).
- Pollard, D. A retrospective look at coupled ice sheet-climate modeling. *Climatic Change* **1** (2010).
- Poulsen, C. J. et al. Did the rifting of the Atlantic Ocean cause the Cretaceous thermal maximum? *Geology* **31** (2003).
- Pucéat, E. et al. Thermal evolution of Cretaceous Tethyan marine waters inferred from oxygen isotope composition of fish tooth enamels. *Paleoceanography* **18** (2003).
- Rose, K. C. et al. Early East Antarctic Ice Sheet growth recorded in the landscape of the Gamburtsev Subglacial Mountains. *Earth and Planetary Science Letters* **375** (2013).
- Sewall, J. O. et al. Climate model boundary conditions for four Cretaceous time slices. *Climate of the Past* **3** (2007).
- Wang, Y. et al. Paleo- CO_2 variation trends and the Cretaceous greenhouse climate. *Earth-Science Reviews* **129** (2014).

Figures

Figure 1. Sensitivity tests to temperature and precipitation forcing fields of the Turonian ice sheet at 560 ppm after 10 kyrs of simulation. (Left) Turonian temperature field and Aptian precipitation field. (Right) Aptian temperature field and Turonian precipitation field.

Figure 2. Statistically significant (at 95%, coloured cells) SST difference between the Turonian and the Aptian (left) and the Turonian and the Maastrichtian (right).

Figure 3. Statistically significant (at 95%, coloured cells) temperature (at 2 m) difference between the Turonian and the Aptian (left) and the Turonian and the Maastrichtian (right).

Figure 4. Mean summer insolation variations over the period 42.5 – 37.5 Ma (after Laskar et al. 2004). The red dashed line denotes the mean summer insolation obtained with our chosen constant orbital parameters.

REVIEWERS' COMMENTS:

Reviewer #1 (Remarks to the Author):

Many thanks to the authors for their comprehensive and complete response to the reviewers' comments. It is very nice when authors actually take up the suggestions from the reviewers rather than just rebutting them, so many thanks for your efforts! I do not have any further substantial comments. Minor comments follow:

Line 17: induces a high resilience *to glaciation*

Line 29 - make clear that values 5 and 15 are anomalies relative to modern, not absolutes.

Line 63 - add co2 values for the new sensitivity studies.

Line 143 - in their response, the authors claim that the important features of the Turonian paleogeography which inhibit glaciation are robust. In the paper itself, they should also make this claim around line 143, and give references to support this.

The paper is written with several phrases of non-standard (mostly french!) phrases. I would expect the Nature copy-editors to correct these, e.g. Line 82 - 'to precise' is not a verb in English; could say 'to further constrain'.

Reviewer #2 (Remarks to the Author):

I was previously reviewer #2. I'm happy that all of my previous comments on the manuscript have been satisfactorily dealt with. The added explanation does make for a more rewarding read and I think that it is now certainly worthy of publication. However, there are a number of improvements needed in the structure and there are numerous grammatical errors. The purpose of the study should be more clearly stated in the introduction, in particular why changes in paleogeography might be an interesting thing to study. The manuscript would benefit from being split into sections. There are a number of minor corrections listed below, these are mostly towards the beginning of the manuscript.

Minor corrections:

L13: The first sentence is overly long, the main point can be made with: "The historical view of a uniformly warm Cretaceous is being increasingly challenged by the accumulation of new data hinting at the possibility of glacial events, including during the Cenomanian-Turonian (~95Ma), the warmest interval of the Cretaceous. "

L18: "high resilience" - not sure what is meant by this. This statement is made much more clearly in the conclusions (L232).

L29: add 'warmer' after 5°C and 15°C.

L29: 'Reduced meridional temperature gradients kept..', suggest changing to 'This led to a reduced meridional temperature gradient, as indicated...'

L27-L31 and L38-41. Could be clearer that the initial sentences are referring to the broad Cretaceous climate, otherwise the discussions of high latitude warmth / cooling appear contradictory. This point is made in the sentence at L31-33, but could be stronger, e.g. 'This broad

view of Cretaceous warmth has been challenged in recent decades by the emergence of data suggesting greater variability in the climate of the Cretaceous".

L43-48. This very long sentence is hard to understand. Suggested changes:

"Arguments in support of Cenomanian-Turonian ice sheets are dependent on interpretation of oxygen isotope records and sequence stratigraphy, although the validity of these proxies as recorders of glacio-eustasy is also debated. The downward revision of atmospheric pCO₂ estimates for the mid-Cretaceous, to levels below the Antarctic glacial threshold suggested by previous modeling studies, adds support to the possibility of Cenomanian-Turonian ice sheets"

L49. In introducing this study it would be preferable to outline why changes in paleogeography may be an interesting thing to study, rather than simply being the first to use a coupled ice sheet model in the Cretaceous. I think that this study is interesting based on its results, not because it is the first to use a particular method.

L55. 'we realise' to 'we perform'.

L63. With added experiments at different CO₂ concentrations it may be best to use a range "..are prescribed (280 to 1120 ppm)..". Alternatively add ", with additional sensitivity experiments at other pCO₂ levels included in the supplement" to the end of the sentence.

L72: add 'In our experiments..' to start of sentence.

L82: 'to precise the', change to 'to more precisely determine'

L83: 'for each of the', to 'for each'

L88: remove 'comprised'

L94: remove '560 ppm' from start of sentence. Suggest changing to "Antarctic ice mass balance analyses for the 560 ppm experiments show..."

L96. 'Antarctica' to 'Antarctic'

L101. As you've stated that accumulation is similar across experiments I think that these additional experiments could be moved to the supplement (I know that you were asked to perform these).

L111. 'in details' to 'Regionally'?

L157. 'an' to 'a'

L166. 'There occurs'?

L169. 'augmentation' to 'increase'

L173. 'enhances amount of solar radiations' to 'enhances the amount of solar radiation'

L203. 'on that the' to 'to the'

L217. 'larger' to 'higher'